# Human Brain Injury and miRNAs: An Experimental Study

**DOI:** 10.3390/ijms20071546

**Published:** 2019-03-27

**Authors:** Francesco Sessa, Francesca Maglietta, Giuseppe Bertozzi, Monica Salerno, Giulio Di Mizio, Giovanni Messina, Angelo Montana, Pietrantonio Ricci, Cristoforo Pomara

**Affiliations:** 1Department of Clinical and Experimental Medicine, University of Foggia, 71122 Foggia, Italy; francesco.sessa@unifg.it (F.S.); maglietta.f@gmail.com (F.M.); gius.brt@gmail.com (G.B.); giovanni.messina@unifg.it (G.M.); 2Department of Medical, Surgical and Advanced Technologies “G.F. Ingrassia”, University of Catania, 95121 Catania, Italy; monica.salerno@unifg.it (M.S.); angelomontana49@gmail.com (A.M.); cristoforo.pomara@unict.it (C.P.); 3Department of Legal, Historical, Economic and Social Sciences, University of Catanzaro, 88100 Catanzaro, Italy; giulio.dimizio@unicz.it; 4Institute of Legal Medicine, University “Magna Graecia” of Catanzaro, 88100 Catanzaro, Italy

**Keywords:** miRNA, miR-21, miR-34, miR-124, miR-132, miR-200b, stroke, aging, cocaine, brain injury

## Abstract

Brain damage is a complex dysfunction that involves a variety of conditions whose pathogenesis involves a number of mediators that lead to clinical sequelae. For this reason, the identification of specific circulating and/or tissue biomarkers which could indicate brain injury is challenging. This experimental study focused on microRNAs (miRNAs), a well-known diagnostic tool both in the clinical setting and in medico-legal investigation. Previous studies demonstrated that specific miRNAs (miR-21, miR-34, miR-124, miR-132, and miR-200b) control important target genes involved in neuronal apoptosis and neuronal stress-induced adaptation. Thus, in this experimental setting, their expression was evaluated in three selected groups of cadavers: drug abusers (cocaine), ischemic-stroke-related deaths, and aging damage in elder people who died from other neurological causes. The results demonstrated that the drug abuser group showed a higher expression of miR-132 and miR-34, suggesting a specific pathway in consumption-induced neurodegeneration. Instead, miR-200b and miR-21 dysregulation was linked to age-related cognitive impairment, and finally, stroke events and consequences were associated with an alteration in miR-200b, miR-21, and miR-124; significantly higher levels of this last expression are strongly sensitive for ischemic damage. Moreover, these results suggest that these expression patterns could be studied in other biological samples (plasma, urine) in subjects with brain injury linked to aging, drug abuse, and stroke to identify reliable biomarkers that could be applied in clinical practice. Further studies with larger samples are needed to confirm these interesting findings.

## 1. Introduction

Brain damage and/or dysfunction as sequelae of different conditions are considered an important field of research for the scientific community. Recent studies have focused on the side effects related to aging on the central nervous system (CNS). Moreover, a growing number of investigations have been conducted, analyzing the effects of stroke and drug use/abuse.

It is well known that in developed countries this kind of problem occurs in a large number of people and is becoming a social and economic problem. The number of older people is constantly increasing, generating a high cost for public health [1]. However, a sedentary lifestyle combined with other factors such as diet and genetic factors can represent the main risk factors for brain stroke [2]. Finally, in the last few years the greater availability and variability of drugs combined with a modern lifestyle have generated a large number of addicts [3,4,5].

Since 2007 the development of microRNA (miRNA) technologies has become an essential part of research projects [6,7,8]. The scientific community has frequently investigated these technologies, considering them as potential molecular biomarkers for several diseases. Today, miRNA dosage has become an essential tool in several clinical applications, such as viral infection diagnosis [9], cancer characterization and prediction of the course of a disease [10,11,12], cardiovascular disorder diagnosis [13,14,15,16], identification of specific patterns in primary muscular disorders [17], and identification of differences between diagnosed type 1 diabetes and healthy controls [18]. Moreover, these biomarkers are becoming very important in brain injury research.

The databases Medline, Cochrane Central, Scopus, Web of Science, Science Direct, EMBASE, and Google Scholar were searched from 2007 to June 2017, using the following keywords: “Brain Injury”, “miRNA dysregulation”, “Stroke”, “Aging”, “Drug Abuser”. The main keywords, “Brain Injury” and “miRNA”, were searched for in association with each of the others.

At the end of the literature review, several miRNAs were selected as potential biomarkers for different brain injuries. Particularly, miRNA hsa-miR-132-3p, hsa-miR-200b-3p, hsa-miR-21-5p, hsa-miR-34a-5p, and hsa-miR-124-5p have been linked with neuronal death, neurogenesis, angiogenesis, platelet aggregation, leukocyte infiltration, blood–brain barrier (BBB) disruption, and tissue infarction [19,20].

This study aimed to identify specific biomarkers (miRNAs) that could be useful in forensic investigations to identify the exact cause of brain injury among several diseases. Particularly, the goal of this experimental study was the identification of specific miRNAs for different kinds of brain damage such as brain stroke, drug use, and aging. 

To achieve this goal, the expression levels of five miRNAs (miRNA hsa-miR-132-3p, hsa-miR-200b-3p, hsa-miR-21-5p, hsa-miR-34a-5p, hsa-miR-124-5p) were chosen after a systematic literature review and evaluated in the brain tissues of 15 selected cases, subdivided into three groups:-stroke group (SG): 5 men who died from brain stroke;-drug group (DG): 5 men who died from drug abuse (cocaine);-aged group (AG): 5 older men who died from Sudden Cardiac arrest.

The expression levels of these five miRNAs were statistically analyzed to evaluate inter-group variations.

## 2. Results

### miRNA Quantitative Real-Time PCR (qRT-PCR) 

Quantitative analysis was used to evaluate the expression levels of miRNA hsa-miR-132-3p, hsa-miR-200b-3p, hsa-miR-21-5p, hsa-miR-34a-5p, and hsa-miR-124-5p, in the three groups SG, DG, and AG. In Table 1, the results of the expression values of each miRNA tested, subdivided for each group, are summarized. 

Moreover, a statistical analysis was performed analyzing the expression values of each miRNA tested, comparing all groups. The expression values of miR-132 are summarized with a box plot analysis in Figure 1.

There were statistically significant differences among group means as determined by one-way ANOVA [F(2,12) = 3.88, *p* < 0.05 (8.09 × 10^−7^)]. Moreover, comparing the expression levels of miRNA-132 singularly, it was significantly higher in the DG group. Indeed, analyzing the data of this group with the others, it showed significantly higher expression levels compared to both the SG group [F(1,8) = 5.31, *p* < 0.05 (4.92 × 10^−5^)] and the AG group [F(1,8) = 5.31, *p* < 0.05 (8.12 × 10^−5^)]. There were no significant differences in the expression levels of miR-132 when comparing the SG and AG groups [F(1,8) = 5.31, *p* = 0.07].

Performing a box plot analysis of the expression levels of miR-200b in each group (Figure 2), we found these were higher in the SG and AG groups compared to in the DG group [F(2,12) = 3.88, *p* < 0.05 (0.005)]. Furthermore, no significant differences were found between the expression values of this miRNA when comparing the SG group and the AG group [F(1,8) = 5.31, *p* = 0.77].

Similar results were obtained when analyzing the expression levels of miR-21 (Figure 3). The expression levels were significantly higher in the SG and AG groups [F(2,12) = 3.88, *p* < 0.05 (0.001)], while no significant differences were found between the expression values of this miRNA when comparing the SG group and the AG group [F(1,8) = 5.31, *p* = 0.47].

Analyzing the expression values of miR-34 (Figure 4), the data showed that this miRNA was significantly more highly expressed in the DG group, with respect to both the SG group [F(1,8) = 5.31, *p* < 0.05 (1.38 × 10^−5^)] and the AG group [F(1,8) = 5.31, *p* < 0.05 (5.8 × 10^−7^)]. No significant differences were found when comparing the SG group with the AG group [F(1,8) = 5.31, *p* = 0.66].

Finally, the data of miR-124 showed a significant difference (Figure 5): expression levels were higher in the SG group with respect to both the DG group [F(1,8) = 5.31, *p* < 0.05 (4 × 10^−8^)] and the AG group [F(1,8) = 5.31, *p* < 0.05 (1.82 × 10^−10^)]. Moreover, the expression values of this miRNA were significantly higher in the DG group with respect to the AG group [F(1,8) = 5.31, *p* < 0.05 (0.03)].

## 3. Discussion

Brain damage is a complex dysfunction that involves a variety of conditions whose pathogenesis includes a number of mediators that lead to clinical sequelae. For this reason, the identification of specific circulating and/or tissue biomarkers which could indicate brain injury is challenging [21,22,23]. Moreover, this is true whether diagnosing early in order to benefit patients by starting appropriate treatment early or whether investigating the cause of death, except for cases in which the circumstances of the event are strongly evocative.

In this context, miRNAs can play an important role in modulating a variety of brain conditions and can serve as new biomarkers. The miRNAs selected in this study (miR-21, miR-34, miR-124, miR-132, and miR-200b), have been predicted to control important target genes involved in neuronal apoptosis and neuronal stress-induced adaptation.

In detail, as well as being a strong indicator of widespread axonal damage [24], miR-21 could be a diagnostic biomarker of cerebral ischemia [25]. The miR-21 pathway of expression seems to be related to the activity of Akt signaling by suppression of phosphatase and tensin homolog (PTEN)-gene expression, resulting in further suppression of caspase-3 expression [26]. This cascade prevents apoptosis of cortical neurons. In addition, patients with stroke and atherosclerosis show significantly higher plasmatic levels of miR-21, which has been interpreted as a strong anti-apoptotic prevention measure [27]. Furthermore, the study conducted by Buller et al., employing in situ hybridization, demonstrated that miR-21 expression was upregulated in neurons in the area adjacent to the ischemic area as an expression of protection from ischemic neuronal death [28]. Finally, miR-21 acts as an anti-inflammatory marker during ischemic stroke [29]. This evidence is in line with our results with miR-21 expression levels being significantly higher in the SG and AG groups compared to the control trauma group and the DG group.

Similar considerations can be made about miR-124. In fact, Jeyaseelan et al. demonstrated that miR-124 was increased in a murine model following cerebral artery occlusion and 24 h reperfusion [30]. Moreover, miR-124 is significantly increased, as well as miR-21, in the area surrounding an ischemic zone. In the same animal model, the nervous overexpression of miR-124 correlated with a decreased infarct size [19]. The probable explanation would result in the anti-apoptotic genes targeted by miR-124 being Bcl-2 and Bcl-xL [31]. According to this data, miR-124 expression, presented in Section 2, showed significant differences, resulting in higher expression in the SG group with respect to the other two groups. A further target of miR-124 is the REST (RE1 Silencing Transcription Factor) gene that determines a reduction of the expression of neuronal plasticity genes (synaptophysin, BDNF, Brain Derived Neurotrophic Factor). Cocaine induces REST and suppresses miR-124, influencing tolerance, sensitization, and addiction [32]. In our experimental setting, the expression values of this miRNA were significantly higher in the DG group with respect to the AG group. Moreover, this neuroprotective function of miR-124 could also be extended to other neurodegenerative diseases. In fact, the miR-124 level in the brain of patients with Alzheimer’s disease is down-regulated, in parallel to the increase in expression of beta-site expression APP-cleaving enzyme 1 (BACE1) [33]. In addition to this mechanism of neurodegeneration, miR-124 is involved in long-term plasticity affecting the transcription factor CREB (cAMP response element-binding protein) [34]. Our experimental findings showed that miR-124 expression in the AG group was similar to that in the control group, showing the lowest statistical significance in inter-group variation. The expression was statistically higher in the SG group, leading us to hypothesize its role as a diagnostic tool in the diagnosis of ischemic stroke. However, cognitive decline can be attributed to complex interactions involving cellular dysfunction, cumulative over time, and life habits that cause the reduction of plasticity in the elderly. The overexpression of miR-34 was related to an increased rate of apoptosis associated with a decrease in Bcl-2 expression in mouse models and in human cell lines [35]. In contrast, an induced down-regulation of miR-34 blocked the inhibition of the expression of the target gene, Bcl-2. In addition, the high levels of the post-transcriptional protein that were induced were accompanied by concomitant low levels of Bax expression and low cleavage by caspases. This mechanism was also observed in the brains of mice with a low-calorie diet. Furthermore, in an experimental study on the hippocampus of rats, in order to evaluate the evolution of neurodegenerative diseases, such as Alzheimer’s disease, miR-34 levels were found to be elevated [36]. Therefore, in our analysis, the expression values of miR-34 were higher in all three groups compared to the control group, probably interpretable as an important index of neurodegeneration being statistically significant in the DG group.

In the SG group, an important role was played by miR-132, whose function is to regulate the glutamate receptor expression level as well as post-stroke excitotoxicity [37]. In fact, some recent studies have shown that the use of miR-132 antagomir during cerebral ischemic attacks may have neuroprotective effects through the suppression of glutamate receptor expression, acting on CREB as stated for miR-124 [38,39]. Moreover, higher serum miR-132 levels were recently suggested as biomarkers for mild cognitive impairment, a stage often preceding Alzheimer’s disease [40]. Considering this effect on memorization processes, recent studies have proved that miR-132 is induced in culture by neurotrophins and neuronal activity and is able to modulate dendritic morphology via the suppression of p250 GTPase-activating protein (p250GAP) [41,42]. Acting between the neural and immune system, miR-132 has recently been discovered to reduce brain inflammation and to increase the level of acetylcholine [43,44,45]. In our experimental setting, the expression levels of miR-132 were higher in all three groups. Therefore, the DG group presented significantly higher levels than the SG and AG groups.

The reason for this could be found in some studies demonstrating the upregulation of mature miR-132 expression in rats following cocaine self-administration (SA). Furthermore, recent observations from Hollander et al. confirmed that 6 h (but not 1 h) access to cocaine SA for 7 days increased miR-132 expression in the rat striatum [46]. The significant increase in miR-132 levels was long lasting and remained high in rats that had been withdrawn from cocaine SA. Changes in dopamine and glutamate neurotransmission, alterations in specific signaling pathways, and/or epigenetic regulation have been advocated as a possible explanation. As chronic cocaine exposure alters dopamine and glutamate signaling by stimulating the dopamine D1 and glutamate NMDA receptors, these targets trigger the activation of the corresponding downstream signaling pathways and lead to CREB-dependent gene expression [47].

Another miRNA expressed in brain tissue, particularly in microglia, and involved in inflammatory response is miR-200b. Oligodendrocytes (OL) are myelin-forming cells of the central nervous system that are vulnerable to cerebral ischemia. The loss of OL and myelin impairs axonal function and is detrimental to functional recovery. Another study by Buller et al. proved that ischemic stroke causes an up-regulation of Serum response factor (SRF) and a down-regulation of miR-200b in OL white matter [48], indicating that miR-200b plays an important role in stroke-induced SRF up-regulation, which ultimately affects OL progenitor cell differentiation. Analogously, microglia treated with miR-200b-inhibitor cause neuronal apoptosis in cell culture that is due to an excessive release of inflammatory cytokines and NO in the conditioned medium via increased cJun activity [49]. In addition, impaired microglial migration has been shown to contribute to the pathogenesis of several brain diseases such as Prion disease [50], Parkinson’s disease [51], and Alzheimer’s disease [52], and it inhibits axonal regeneration during acute CNS injury [53]. However, further experiments are required to ascertain the possible neuroprotective role of microglial miR-200b. These literature data are in line with the statistical results of our experimental setting: the expression levels of miRNA-21 and miRNA-200b were higher in the SG and AG groups compared to the control group, while in the DG group, the expression was similar to that in the control group.

## 4. Material and Methods

### 4.1. Selected Cases

Cases were selected after analyzing the documentation of all autopsies performed by the Institute of Legal Medicine of Foggia from 2001 to 2018 (about 1500 autopsies). Five cases of men who died from brain stroke were selected (mean age 57.8 ± 6.7 years) and made up the SG group; five cases of men who died with a toxicological test positive for drug abuse (cocaine) were selected (mean age 29.2 ± 5.6 years) and made up the DG group; and five cases of older men who died from Sudden Cardiac arrest (mean age 75.8 ± 6.9 years) made up the AG group.

Finally, two cases of previously healthy men (mean age 41 ± 1.5 years) who died in car accidents from causes other than brain trauma were selected as controls in the Real-Time PCR reactions.

All procedures were performed in accordance with international guidelines and were approved by the Scientific Committee of University of Foggia (Italy–FGBJ_23/10/2018).

The selected groups are summarized in Table 2:

### 4.2. miRNA Quantitative Real-Time PCR (qRT-PCR)

Total RNA, including miRNAs, was isolated from formaldehyde-fixed paraffin-embedded (FFPE) samples (four 20 µm sections) using the Recover All Total Nucleic Acid Isolation Kit (Applied Biosystems Life Technologies, Foster City, CA, USA) with minor modifications. Briefly, before RNA extraction, all samples were deparaffinated and processed as indicated in the protocol. Finally, the purified RNA was eluted with 65 μL RNAse-free water. Quantification was performed in heart and musculoskeletal tissues.

The quantification of RNA was performed with the Qubit RNA HS Assay Kit (Applied Biosystems Life Technologies, Foster City, CA, USA) using the Qubit Fluorometer; it provides an accurate and selective method for the quantitation of low-abundance RNA samples.

For miRNA profiling, the TaqMan Advanced miRNA Assay (Applied Biosystems, Darmstadt, Germany) was used. This kit is composed of pre-formulated primer and probe sets that are designed for the analysis of miRNA expression levels. The assays were selected at thermofisher.com/advancedmirna.

The miRNAs tested in the selected samples are summarized in Table 3:

cDNA was obtained following the TaqMan Advanced miRNA Assays User Guide (Applied Biosystems, Publication number 100027897 Rev. C). Quantitative real-time PCR (qRT-PCR) was performed using the StepOnePlus Real-Time PCR System (Applied Biosystems Darmstadt, Germany), and raw data were analyzed using the accompanying software (version 2.3). As described in the user’s guide for the Fast reaction plate, the qRT-PCR reactions (in triplicate) were performed with incubation at 95 °C for 20 s, followed by 40 cycles of 95 °C for 1 s and 60 °C for 20 s. A negative control without cDNA was also included in parallel.

The data obtained were compared with the data from endogenous controls; as described in the user guide, they showed that gene expression was relatively constant across tissues. Normalization to endogenous control genes is currently the most accurate method to correct for potential biases that are caused by sample collection, variation in the amount of starting material, RT efficiency, and RNA preparation and quality. For this experiment, “has-miR-186-5p” (TaqMan Advanced miRNA Assays, Applied Biosystems Life Technologies, Foster City, CA, USA) was used as an endogenous control (Table 2).

Expression fold changes were computed using the 2^−ΔΔCt^ calculation [54], where ∆Ct = Ct(test miRNA) − Ct(mir-186-5p) and ∆∆Ct = ∆Ct(individual sample) − ∆Ct(control median samples, Table 2).

### 4.3. Statistics

Unless specified otherwise, data are expressed as means ± SEM versus baseline. Statistical comparisons were performed using one-way ANOVA (Microsoft Office Excel 2007) and post hoc pair wise comparisons were performed using Tukey’s Honestly Significant Difference (the TukeyHSD function in R). A value of *p* < 0.05 was considered a statistically significant difference. Descriptive and statistical comparisons was performed using Microsoft Office Excel 2007.

## 5. Conclusions

Although brain injury results from complex pathophysiological interactions, miRNAs are considered reliable biomarkers. In our experimental setting, drug (cocaine) consumption was found to be related to higher expression of miR-132 and miR-34, suggesting a specific pathway in its induced neurodegeneration. Instead, miR200b and miR-21 dysregulation should be linked to age-related cognitive impairment. Finally, stroke events and consequences could be associated with alterations in miR-200b, miR-21, and miR-124, with significantly higher levels of miR-124 expression being very sensitive for ischemic damage.

Studies with larger samples are mandatory to confirm our interesting findings with the aim of using these molecules as biomarkers in forensic investigation: the higher expression of miR-132 and miR-34 may be used as a marker for brain injury from drug (cocaine) consumption; the expression pattern of higher levels of miR-200b and miR-21 and the threshold value of mir-124 expression could indicate cases of age-related cognitive impairment; and the expression pattern of higher levels of miR-200b, miR-21, and miR-124 is indicative of ischemic brain injury. These expression patterns could be very useful as easily available tissue biomarkers in medico-legal investigations to ascertain the exact cause of death in suspected brain injury cases.

On the other hand, our results suggest that these expression patterns could be studied in other biological samples (such as plasma and urine) in living subjects with brain injury linked to aging, drug abuse, and stroke. The identification of new molecular biomarkers could be very useful in clinical practice to quickly orient diagnosis and determine the most suitable treatment in the shortest time. Further studies should be carried out with this specific purpose.

## Figures and Tables

**Figure 1 ijms-20-01546-f001:**
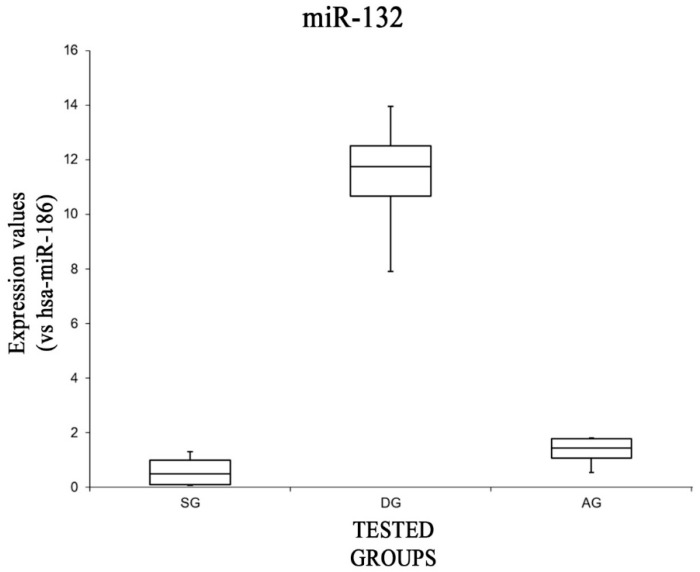
Box plot analysis comparing expression levels of miR-132 (endogenous control miR-186) in each group.

**Figure 2 ijms-20-01546-f002:**
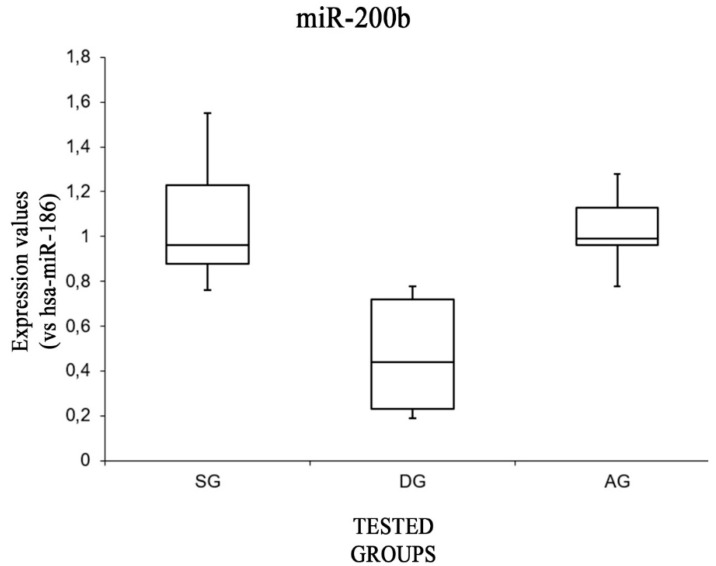
Box plot analysis showing the expression levels of miR-200b (endogenous control miR-186) in each group.

**Figure 3 ijms-20-01546-f003:**
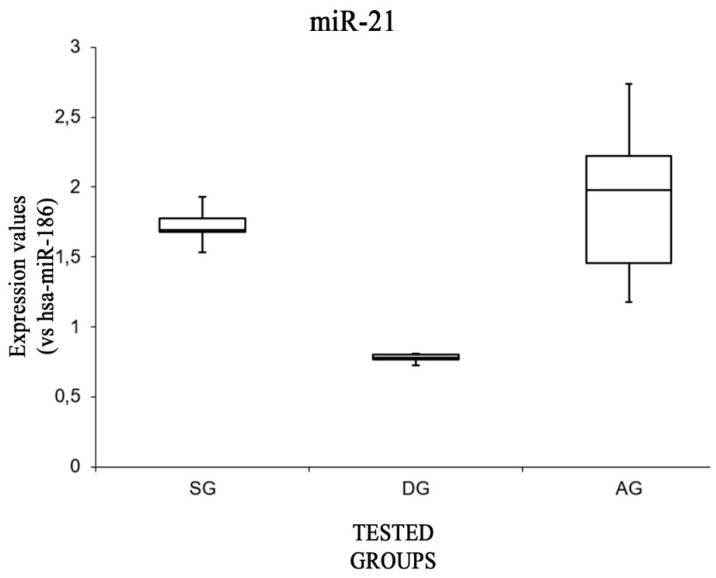
Box plot analysis showing the expression levels of miR-21 (endogenous control miR-186) in each group.

**Figure 4 ijms-20-01546-f004:**
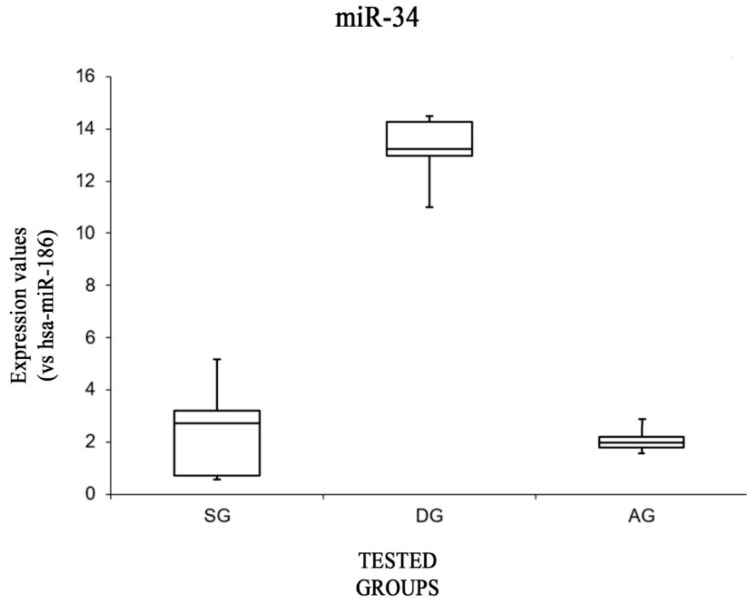
Box plot analysis showing the expression levels of miRNA 34 (endogenous control miR-186) in each group.

**Figure 5 ijms-20-01546-f005:**
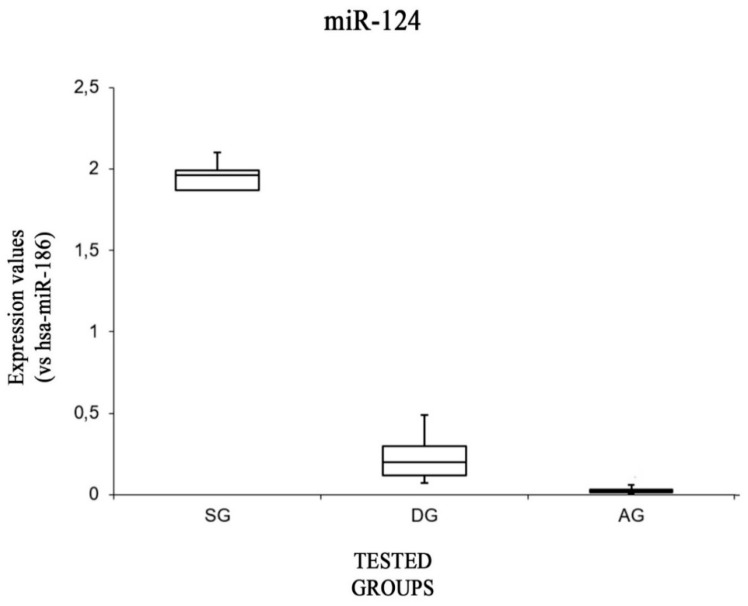
Box plot analysis showing the expression levels of miR-124 (endogenous control miR-186) in each group.

**Table 1 ijms-20-01546-t001:** Expression levels of microRNAs (miRNAs) analyzed in each group, reporting the mean values of ∆CT (threshold cycle) and ∆∆CT. SG, stroke group; DG, drug group; AG, aged group.

Mean Expression Levels (Endogenous Control: miR-186)						
Groups		miR-132	miR-200b	miR-21	miR-34	miR-124
**SG**	**∆CT (Mean Values)**	3.93 ± 1.98	−0.52 ± 0.40	−2.06 ± 0.20	3.81 ± 1.40	−1.40 ± 0.07
	**∆∆CT (Mean Values)**	1.47 ± 1.98	−0.06± 0.40	−0.74 ± 0.20	−0.84 ± 1.40	−0.96 ± 0.07
	**Fold Difference = 2^−∆∆CT^ (Mean Values)**	0.62 ± 0.53	1.07 ± 0.31	1.69 ± 0.23	2.47 ± 1.91	1.95 ± 0.09
**DG**	**∆CT (Mean Values)**	2.17 ± 0.44	0.81 ± 0.91	−0.92 ± 0.13	0.95 ± 0.19	1.91 ± 1.05
	**∆∆CT (Mean Values)**	−3.41 ± 0.44	1.27 ± 0.91	0.38 ± 0.13	−3.69 ± 0.19	2.35 ± 1.05
	**Fold Difference = 2^−∆∆CT^ (Mean Values)**	11.042 ± 2.91	0.47 ± 0.27	0.76 ± 0.06	13.06 ± 1.65	0.23 ± 0.16
**AG**	**∆CT (Mean Values)**	2.16 ± 0.72	−0.49 ± 0.26	−2.16 ± 0.48	3.63 ± 0.33	5.42 ± 0.73
	**∆∆CT (Mean Values)**	−0.29 ± 0.72	−0.02 ± 0.26	−0.95 ± 0.48	−0.76 ± 0.33	5.86 ± 0.73
	**Fold Difference = 2^−∆∆CT^ (Mean Values)**	1.33 ± 0.53	1.02 ± 0.18	1.91 ± 0.61	2.07 ± 0.5	0.04 ± 0.05

**Table 2 ijms-20-01546-t002:** Selected groups: SG, stroke group; DG, drug group; AG, aged group; CTR, control group.

Groups	Age	Cause of Death	Macroscopic Brain Findings	Microscopic Brain Findings
**SG**				
**1**	80	ischemic stroke	brain edema; reddish punctiform areas of the white matter	perineuronal and perivasal edema, red neurons, malacic areas
**2**	79	ischemic stroke from a vertebrobasilar embolism	brain edema; reddish punctiform areas of the white matter	perineuronal edema, red neurons
**3**	34	ischemic stroke in a subject with atheromatous formation at the Willis polygon	atherosclerotic alterations in the Willis polygon.	vasogenic edema, red neurons
**4**	24	ischemic stroke from a systemic massive bleeding	post-hypoxic malacic areas of the white matter	perivasal edema, red neurons, hemorrhages of Duret.
**5**	72	ischemic stroke	brain edema; reddish punctiform areas of the white matter	vasogenic edema, ischemic areas, “red neurons”
**DG**				
**1**	29	sudden cardiac death secondary to cocaine intake	brain edema	cortical edema
**2**	32	fatal ventricular arrhythmia secondary to cocaine intake	brain edema	vasogenic edema
**3**	30	sudden cardiac death secondary to cocaine intake	brain edema	perineuronal and perivasal edema
**4**	20	fatal cardiac arrhythmia secondary to cocaine intake	brain edema	cortical edema, arteriolosclerosis
**5**	35	fatal ventricular arrhythmia secondary to cocaine intake	brain edema, stasis	perineuronal edema, small perivascular hemorrhages
**AG**				
**1**	80	fatal cardiac arrhythmia	brain edema	vasogenic edema, arteriolosclerosis
**2**	83	ventricular arrhythmia	brain edema	perineuronal and perivasal edema
**3**	77	cardiac failure	brain edema	vasogenic edema, small periventricular hemorrhages
**4**	65	sudden cardiac death	brain edema	cortical edema
**5**	74	fatal cardiac arrhythmia	brain edema	perineuronal and perivasal edema, small perivascular hemorrhages
**CTR**				
**1**	27	acute cardiorespiratory arrest in subjects with aortic lesions	brain edema	perineuronal edema, stasis
**2**	55	acute cardiorespiratory arrest in subjects with multiple costal fracture and pulmonary contusions	brain edema	perineuronal and perivasal edema, small perivascular hemorrhages

**Table 3 ijms-20-01546-t003:** Tested miRNAs in this study.

Assay Name	Mature miRNA Sequence:	Chromosome Location
hsa-miR-132-3p	UAACAGUCUACAGCCAUGGUCG	Chr. 17-2049908-2050008
hsa-miR-200b-3p	UAAUACUGCCUGGUAAUGAUGA	Chr.1: 1167104-1167198
hsa-miR-21-5p	UAGCUUAUCAGACUGAUGUUGA	Chr.17: 59841266-59841337
hsa-miR-34a-5p	UGGCAGUGUCUUAGCUGGUUGU	Chr.1: 9151668-9151777
hsa-miR-124-5p	CGUGUUCACAGCGGACCUUGAU	Chr.8: 9903388-9903472
**Endogenous Control Genes**		
hsa-miR-186-5p	CAAAGAAUUCUCCUUUUGGGCU	Chr.1: 71067631-71067716

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
