# Peer review of "Human Brain Injury and miRNAs: An Experimental Study"

_ijms, 2019, doi:10.3390/ijms20071546_

Round 1

Reviewer 1 Report

The knowledge about MicroRNA is evolving, miRNA acts as a versatile regulator of brain development and its functions as well as in the response to insults. we need to improve our understanding about the interactions with clinical situations, CT and MRI features and treatment. This review contributes to establish the current implications of miRNAs in cerebrovascular physiopathology, particularly in three groups patients, cocaine consumption, cognitive impairment snd ischemic stroke.

Author Response

We thank the referees for their interest in our work and for their helpful comments that will greatly improve the manuscript (code: ijms-461231). The referees have brought up some good points and we appreciate the opportunity to clarify our research objectives and results.

We have extensively checked the English, reviewing the manuscript.

The knowledge about MicroRNA is evolving, miRNA acts as a versatile regulator of brain development and its functions as well as in the response to insults. we need to improve our understanding about the interactions with clinical situations, CT and MRI features and treatment. This review contributes to establish the current implications of miRNAs in cerebrovascular physiopathology, particularly in three groups patients, cocaine consumption, cognitive impairment and ischemic stroke.

We have reviewed the manuscript, improving the research design, the results and discussion sections.

Reviewer 2 Report

1) title is misleading and do not reflect the objective.

2) objective is not SMART. A more tangible objective is needed.

3) the author performed systematic review in the first place but no results is shown. Then the author continued with the so-called experimental study. Yet, what are the research gap, hypothesis?

4) the cycling condition of qpcr was not mentioned.

5) what are the reference genes? what is your controls for qpcr?

6) a better description of statistical analysis is needed. Why 4 'statistical software' are needed. There is merely a simple study, a straight forward analysis will do.

7) What are the reference group you used to generate the results?

8) any duplicate/triplicate has been done? 

9) your delta ct and delta delta ct value are? 

10) If there is no duplicate/triplicate, i don't think the results is convincing.

11) a more solid discussion is needed.

12) funding was not acknowledge.

13) Quite a number of grammatical errors detected.

14) Several technical errors also detected.

pg 1, line 37 etc. need to be specific

15) recheck the format/spacing/grammatical

projects[6-8].

them as

identifying possible

...many more

16) this study involved human samples, yet, ethical consent was not mentioned.

Author Response

We thank the referees for their interest in our work and for their helpful comments that will greatly improve the manuscript (code: ijms-461231). The referees have brought up some good points and we appreciate the opportunity to clarify our research objectives and results.

Reviewer #2:

We have improved the manuscript.

Comments and Suggestions for Authors

1)                   title is misleading and do not reflect the objective.

We changed the title as follows: “Human brain injury and miRNAs: an experimental study”

2)                   objective is not SMART. A more tangible objective is needed.

We clarified the aims and the methods, by means of which our aims were reached.

3) the author performed systematic review in the first place but no results is shown. Then the author continued with the so-called experimental study. Yet, what are the research gap, hypothesis?

Our hypothesis is described in the introduction section and thanks to our statistical analyses a hypothesis for further studies is also suggested in the conclusion section.

4) the cycling condition of qpcr was not mentioned.

We have reviewed the Material and Methods section.

5) what are the reference genes? what is your controls for qpcr?

We have reviewed the Material and Methods section.

6) a better description of statistical analysis is needed. Why 4 'statistical software' are needed. There is merely a simple study, a straight forward analysis will do.

We have reviewed the Material and Methods section.

7) What are the reference group you used to generate the results?

We have reviewed the Material and Methods section.

8) any duplicate/triplicate has been done? 

We have reviewed the Material and Methods section.

9) your delta ct and delta delta ct value are? 

We have reviewed table 1, inserting the requested data.

10) If there is no duplicate/triplicate, i don't think the results is convincing.

We have reviewed the Material and Methods section.

11) a more solid discussion is needed.

We improved the discussion section clarifying the results of our experimental setting compared to the systematic literature review on the topic.

12) funding was not acknowledge.

We added this information.

13) Quite a number of grammatical errors detected.

We checked grammatical errors and reviewed the text with the help of the Scientific Bureau of the University of Catania for language support.

14) Several technical errors also detected.

pg 1, line 37 etc. need to be specific

We checked technical errors.

15) recheck the format/spacing/grammatical projects[6-8].

them as

identifying possible

...many more

We checked these typos.

16) this study involved human samples, yet, ethical consent was not mentioned.

We mentioned it in Selected cases in the M&M section.

Please do not hesitate to contact me for any further questions.

Pietrantonio Ricci, MD, PhD

e mail: [email protected] .it

Tel: (39) 0881 736900

Full Professor,

Department of Clinical and Experimental Medicine,

University of Foggia, Foggia, Italy

Viale degli Aviatori 1

71121 Foggia, Italy

Round 2

Reviewer 2 Report

Please list down the changes made in the point by point response, page, paragraph and line number, to avoid confusion and increase the chances of acceptances.

Author Response

Reviewer #2:

Open Review

English language and style

(x) Extensive editing of English language and style required 
( ) Moderate English changes required 
( ) English language and style are fine/minor spell check required 
( ) I don't feel qualified to judge about the English language and style 

We have extensively checked the English, reviewing the manuscript.

Yes

Can be improved

Must be improved

Not applicable

Does the   introduction provide sufficient background and include all relevant   references?

( )

( )

(x)

( )

Is the research   design appropriate?

( )

( )

(x)

( )

Are the methods   adequately described?

( )

( )

(x)

( )

Are the results   clearly presented?

( )

( )

(x)

( )

Are the   conclusions supported by the results?

( )

( )

(x)

( )

We have improved the manuscript.

Comments and Suggestions for Authors

1)     title is misleading and do not reflect the objective.

We changed the title as follows: “Human brain injury and miRNAs: an experimental study”

Page 1, lines 2-3

2)     objective is not SMART. A more tangible objective is needed.

We clarified the aims and the methods, by means of which our aims were reached.

Page 2, lines 68-82.

3) the author performed systematic review in the first place but no results is shown. Then the author continued with the so-called experimental study. Yet, what are the research gap, hypothesis?

Our hypothesis is described in the introduction section and thanks to our statistical analyses a hypothesis for further studies is also suggested in the conclusion section.

Page 2, lines 68-82.

Page 12, lines 298-308

4) the cycling condition of qpcr was not mentioned.

We have reviewed the Material and Methods section.

Page 12, lines 267-272

5) what are the reference genes? what is your controls for qpcr?

We have reviewed the Material and Methods section.

Page 12, lines 279-281.

6) a better description of statistical analysis is needed. Why 4 'statistical software' are needed. There is merely a simple study, a straight forward analysis will do.

We have reviewed the Material and Methods section.

Page 12, line 284; line 288.

7) What are the reference group you used to generate the results?

We have reviewed the Material and Methods section.

Page 12, lines 279-281

8) any duplicate/triplicate has been done? 

We have reviewed the Material and Methods section.

Page 12 line 270.

9) your delta ct and delta delta ct value are? 

We have reviewed table 1, inserting the requested data.

10) If there is no duplicate/triplicate, i don't think the results is convincing.

We have reviewed the Material and Methods section.

Page 12, line 270.

11) a more solid discussion is needed.

We improved the discussion section clarifying the results of our experimental setting compared to the systematic literature review on the topic.

Page 8, lines 155-157

Page 9, lines 176-179

Page 10, line 230-234

Conclusion

Page 12, lines 298-308

12) funding was not acknowledge.

We added this information.

Page 13

13) Quite a number of grammatical errors detected.

We checked grammatical errors and reviewed the text with the help of the Scientific Bureau of the University of Catania for language support.

14) Several technical errors also detected.

pg 1, line 37 etc. need to be specific

We checked technical errors.

15) recheck the format/spacing/grammatical

projects[6-8].

them as

identifying possible

...many more

We checked these typos.

16) this study involved human samples, yet, ethical consent was not mentioned.

We mentioned it in Selected cases in the M&M section.

Page 10, line 247-248

Please do not hesitate to contact me for any further questions.

Pietrantonio Ricci, MD, PhD

Round 3

Reviewer 2 Report

A clear objective is needed. The objective is not SMART.

Author Response

We thank the referee for his interest in our work and for his helpful comments that will improve the manuscript (code: ijms-461231). The referee has brought up some good points and we appreciate the opportunity to clarify our research objectives as stated below.

Reviewer #2:

Open Review

English language and style

( ) Extensive editing of English language and style required 
(x) Moderate English changes required 
( ) English language and style are fine/minor spell check required 
( ) I don't feel qualified to judge about the English language and style 

We have extensively checked the English, reviewing the manuscript with the help of the Scientific Bureau of the University of Catania.

  Yes

Can be   improved

Must be   improved

Not   applicable

Does the   introduction provide sufficient background and include all relevant   references?

(x)

( )

( )

( )

Is the   research design appropriate?

( )

( )

(x)

( )

Are the   methods adequately described?

(x)

( )

( )

( )

Are the   results clearly presented?

(x)

( )

( )

( )

Are the   conclusions supported by the results?

(x)

( )

( )

( )

We have improved the aims section.

Comments and Suggestions for Authors

1)     A clear objective is needed.The objective is not SMART.

We further clarified the aims in the introduction section and commented on it in the conclusion, modifying the abstract.

Page 1, lines 34-36;

Page 2, lines 69-80;

Page 12, lines 315-317.

Please do not hesitate to contact me for any further questions.

Pietrantonio Ricci, MD, PhD